# Feeding Diets Moderate in Physically Effective Fibre Alters Eating and Feed Sorting Patterns without Improving Ruminal pH, but Impaired Liver Health in Dairy Cows

**DOI:** 10.3390/ani9040128

**Published:** 2019-03-29

**Authors:** Iris Kröger, Elke Humer, Viktoria Neubauer, Nicole Reisinger, Qendrim Zebeli

**Affiliations:** 1Institute of Animal Nutrition and Functional Plant Compounds, Department for Farm Animals and Veterinary Public Health, University of Veterinary Medicine Vienna, 1210 Vienna, Austria; iris_kroeger@gmx.de (I.K.); elke.humer@vetmeduni.ac.at (E.H.); viktoria.neubauer@vetmeduni.ac.at (V.N.); 2Institute of Milk Hygiene, Milk Technology and Food Science, Department for Farm Animals and Veterinary Public Health, University of Veterinary Medicine Vienna, 1210 Vienna, Austria; 3BIOMIN Research Center, Biomin Holding GmbH, 3430 Tulln, Austria; nicole.reisinger@biomin.net

**Keywords:** chewing activity, metabolic parameters, milk production, sorting behaviour, physically effective fibre

## Abstract

**Simple Summary:**

This study tested the hypothesis that the content of physically effective fibre in the diet modifies eating and feed sorting patterns of cows, which is reflected in rumen and systemic health variables. We observed that switching cows from a diet high to a diet moderate in physically effective fibre (17.8%; 28% starch on dry matter basis) in an attempt to stabilize rumen pH variations altered eating, sorting, and chewing behaviour. However, this attempt did not prevent ruminal pH drop and the impairment of liver health variables. Therefore, our study emphasizes the importance of increasing the level of physically effective fibre for particles >8 mm beyond 17%–18% of the diet when (barley grain-based) starch content is at a 28% level.

**Abstract:**

The main challenge in dairy cattle feeding is to find a balance between the energy and physically effective fibre (peNDF), required to maintain rumen health. In an attempt to regulate the balance between energy intake and rumen buffering, we hypothesized that the content of peNDF in the diet modifies eating and feed sorting patterns of the cows. Sixteen lactating Simmental cows were switched from a diet high in peNDF, with which they were fed for one week, to a diet moderate in peNDF for four weeks. Data showed that during the moderate peNDF feeding the cows increased sorting for medium-sized particles and avoided both long and very fine particles. In addition, cows decreased their eating time per meal, but increased the number of meals per day, obviously attempting to decrease the amount of fermentable substrate per time unit while maintaining high levels of nutrient/energy intake. Although these changes during the moderate peNDF feeding went along with a lower diurnal variation of ruminal pH, feeding of the latter diet did not prevent ruminal pH drop and increased the level of all liver enzymes, indicating liver tissue damage. In conclusion, the altered eating, chewing, and sorting behaviour of the cows during the moderate peNDF feeding could not alleviate the deficiency in peNDF, which resulted in ruminal pH depression and impairment of liver health variables.

## 1. Introduction

Dairy cows require sufficient amounts of physically effective fibre (peNDF) in their diet to stimulate chewing, promote rumen buffering, optimize feed degradability, and maintain physiological ruminal pH [1]. However, the commercial lactation diets of cows typically contain large amounts of concentrates to enhance nutrient intake, making them low or moderate in peNDF [2]. This is a key challenge in modern dairy cattle nutrition, as feeding insufficient peNDF to dairy cows increases the risk of ruminal fermentation disorders, thus paving the way to subacute ruminal acidosis (SARA) and its subsequent disorders [3].

Another common concern of feeding management in commercial dairy herds is feed sorting, which can jeopardize the prediction of peNDF adequacy of the fed diet [2]. Current recommendations of peNDF in dairy cattle consider peNDF level, the starch content of the diet, and the feed intake of the cows [4], but not the implication of sorting behaviour. Although cows typically sort for concentrate, lowering their peNDF intake, cows may alter their sorting and chewing behaviour during SARA towards an increased sorting for longer particles and against shorter particles in the diet to mitigate short bouts of acidosis [5]. Previous studies have investigated sorting and/or chewing behaviour when cows were exposed to short-term [6,7] or intermittent concentrate challenges [8]. However, if cows alter their eating behaviour and sorting behaviour over time when a total mixed ration (TMR) diet with moderate peNDF level is fed remains unclear, as well as whether or not this event leads to changes in chewing behaviour and lowers diurnal ruminal pH variation.

In addition, more knowledge is required regarding the paraclinical parameters that best predict the degree of peNDF deficiency and occurrence of rumen health disorders [2], along with typical indicators such as ruminal pH and chewing responses. Thus, milk parameters are commonly used as indicators for dietary changes in dairy farms, while liver-associated variables may increase in the blood when toxins such as lipopolysaccharides (LPS) are translocated into systemic circulation during SARA [9]. However, whether or not blood and milk variables can serve as indirect parameters to identify cows suffering from SARA remains unclear.

In this study, we primarily tested the hypothesis that the decrease of peNDF content in the diet, due to a decreased forage to concentrate ratio, modifies eating and feed sorting patterns of cows over time in an attempt to regulate the balance between energy intake and rumen buffering. Another aim of this study was to investigate the cow responses in feed intake, milk variables, chewing indices, blood minerals, and metabolites, as well as liver enzymes, in an effort establish their potential as rumen health indicators in dairy cattle.

## 2. Materials and Methods

All procedures involving animal handling and treatment were approved by the institutional ethics committee of the University of Veterinary Medicine (Vetmeduni), Vienna, and the national authority according to §26 of the Law for Animal Experiments, Tierversuchsgesetz (TVG), 2012 (BMWFW-68.205/0098-WF/V/3b/2016).

### 2.1. Animals and Feeding

The feeding experiment was conducted with 16 lactating multiparous Simmental cows (738 ± 85 kg BW; 35.6 ± 4.7 kg milk per day; 3.6 ± 1.7 lactations, 90.9 ± 22 days in milk) that were kept in a loose-housing stable at the research dairy farm of Vetmeduni Vienna (Pottenstein, Austria). Cows were adapted to a diet with maximum 40% concentrate before the start of the experiment. During the first week of the experiment cows were fed a TMR high in peNDF (HS) (24.4% peNDF > 8 mm and 18.8% starch on dry matter (DM) basis) containing 60% forage and 40% concentrate (DM basis). Afterwards, cows were switched to a diet with 60% concentrate, considered moderate in peNDF (MS; 17.6% peNDF > 8 mm and 27.8% starch on DM basis) as 18.5% peNDF > 8 mm in cows’ diets are recommended to prevent drops in ruminal pH [1,4]. Barley grain was used as a starch source due to its high acidogenic value [1]. The MS diet was fed for four weeks (MS wk 1–4). The composition of the TMRs is shown in Table 1.

The TMR was prepared with an automatic feeding system (Trioliet Triomatic T15, Oldenzaal, The Netherlands) and fresh TMR was offered daily at 0730 and 1730. The diets were offered ad libitum, allowing approximately 10% feed refusals, to meet the voluntary feed intake of the cows. The individual feed intake was continuously recorded using feeding troughs equipped with electronic weighing scales and computer-regulated access gates (Insentec B.V., Marknesse, The Netherlands). Cows had free access to water and a salt lick stone.

### 2.2. Feed Sampling and Analysis

Feed samples of fresh TMR were collected weekly and frozen at −20 °C until they were further analysed. Before analyses, samples were dried at 65 °C in a forced-air oven for 48 h and ground to pass through a 0.5-mm screen (Ultra Centrifugal Mill ZM 200, Retsch, Haan, Germany). Dry matter was determined by oven drying at 100 °C for 24 h. Ash was analysed by combustion in a muffle furnace overnight at 580 °C. Crude protein (CP) was analysed following the Kjeldahl method (CP = N × 6.25) and ether extract (EE) using the soxhlet extraction system (Extraction System B-811, Büchi, Flawil, Switzerland). The neutral detergent fibre (aNDFom) and acid detergent fibre (ADFom) were determined with sodium sulphite and reported exclusive of residual ash following the official analytical methods of VDLUFA [12] using a Fibertherm FT 12 (Gerhardt GmbH & Co. KG, Königswinter, Germany) with heat-stable α-amylase in case of aNDFom. Total starch content was determined using a commercially available kit (Megazyme, Wicklow, Ireland). 

### 2.3. Measurement of Particle Size and Sorting Behaviour

To determine the physical structure of the diet (fibre, particle size, and peNDF) feed samples were taken during HS two days before switching cows to the MS diet (d −2), during MS wk 1 (d 2) and MS wk 4 (d 26). Particle size distribution of the fresh TMR was determined in duplicate using a Penn State Particle Separator with three sieves separating the samples into four particle size fractions: long (>19 mm), medium (8–19 mm), fine (1.18–8 mm), and very fine (<1.18 mm; Pan). The particle distribution of the diets is summarized in Table 1. The physical effectiveness factors (pef; i.e., the cumulative proportion of feed DM retained on sieves of the Penn State Particle Separator) were designated as pef > 8 and pef > 1.18, respectively. The physically effective NDF of either two (peNDF > 8) or three sieves (peNDF > 1.18) were calculated by multiplying the fraction of pef > 8 and pef > 1.18 with the aNDFom content as described previously by Lammers et al. [13].

To determine the feed sorting of each cow, samples of feed refusals were taken during HS (d −1), MS wk 1 (d 3), and MS wk 4 (d 27), before morning feeding. Samples of fresh feed and feed refusals were taken from six locations (2 × bottom, 2 × middle, and 2 × top) of each feeding trough. Particle size distribution of each sample was determined in duplicate. As suggested by Leonardi and Armentano [14], the feed sorting index (SI) was calculated as the actual intake of each particle fraction (>19 mm, 8–19 mm, 1.18–8 mm, and Pan) expressed as the percentage of the predicted intake of that fraction. In detail, the actual intake of each fraction was calculated as the difference between the amount of each fraction in the offered feed and that of the refused feed. The predicted intake of each fraction was calculated as the product of the intake of the total diet multiplied by the fraction in the offered diet in %. A sorting index of 100 indicates no sorting, while an index >100 indicates sorting for particles, and an index <100 implies sorting against particles [14].

### 2.4. Measurement of Chewing Activity

Chewing activity was measured using noseband-sensor halters (RumiWatch System, ITIN + Hoch GmbH, Liestal, Switzerland). Chewing data were processed using RumiWatch Manager 2 (V.2.1.0.0) and RumiWatch Converter (V.0.7.3.2). Measurements were conducted during the last four days of HS (d −3 to 0), during MS wk 1 (d 1–6), and on three consecutive days of MS wk 4 (d 25–27). 

The measurement of ruminating activity and evaluation of the noseband-sensor halters have been reported in detail by Kröger et al. [15]. Chewing data included the duration of eating, ruminating, and total chewing in minutes per day (min/d), the number of ruminating boli (n/d), the chews per bolus (n/bolus), and the chews per minute (n/min). Chewing and feed intake data of the same day were used for calculating the chewing indices (i.e., chews/g of dry matter intake (DMI), min/kg DMI, min/kg total NDF intake, and min/kg peNDF intake) for each chewing category per feeding phase. Ruminating and eating by the cows, based on 10-min intervals (*n* = 4637), served for investigating the diurnal variation of eating and ruminating expressed in minutes per hour (min/h). 

### 2.5. Measurement of the Reticuloruminal pH and Monitoring of SARA Conditions

Before the beginning of the experiment, each cow received an indwelling pH sensor (smaXtec, Graz, Austria) for continuous pH measurement during the whole experiment. After calibration in pH 7.0, the sensors were administered to the cows orally following the instruction protocol of the company. The sensors measured pH every 10 min with an accuracy of ±0.2 units, and transmitted the data in real time to a base station installed in the barn. The pH data served for calculating the pH < 6.0 (min/d), as well as for calculating the minimum, mean, and maximum pH (per d of the experiment). We considered a threshold of reticuloruminal pH < 6.0 for longer than 5–6 h/d to represent SARA conditions, since the common SARA threshold of pH 5.8 in ruminal pH [16] corresponds to a pH of 6.0 when indwelling pH sensors staying in the reticulum are used [17]. Based on the pH data available in 10-min intervals, the mean pH per h was calculated for the days when chewing activity was measured. The pH of 4654 10-min intervals served for calculating the diurnal variation of the mean pH.

### 2.6. Determination of Milk Yield, Milk Composition, and Feed Efficiency

The cows were milked twice daily at 0700 and 1700 in a tandem milking parlour (AlproTM-Milking, DeLaval Inc, Kansas City, MO, USA). Milk yield was registered using an electronic machine recorder (DeLaval Corp, Tumba, Sweden). Cows were enrolled in the study after performing a California mastitis test (DeLaval Corp, Tumba, Sweden) to ensure clinically healthy udders (somatic cell count (SCC) < 100,000). Milk samples were taken on d 0 and 1 (HS), 12 and 13 (MS wk 2), 20 and 21 (MS wk 3), and 28 and 29 (MS wk 4). Milk samples from the afternoon milking and morning milking of the next day were pooled and analysed on SCC, protein, fat, lactose, milk urea nitrogen (MUN), pH, and nonfat dry milk (NDM) by Combifoss (Foss, Hillerød, Denmark). The calculation of energy-corrected milk (ECM) followed the methodology of the German Society of Nutrition Physiology (GfE) [10]: ECM = (0.38 × milk fat% + 0.21 × milk protein% + 0.95) × kg of milk/3.2. The feed efficiency was calculated by dividing the ECM by the DMI on the milk sampling days (kg/kg).

### 2.7. Blood Sampling and Analyses

Blood samples were collected on d 0 (HS), 12 (MS wk 2), 20 (MS wk 3), and 28 (MS wk 4) before morning feeding. The samples were taken from the jugular vein using 10 mL evacuated tubes (Vacuette; Greiner Bio-One GmbH, Kremsmünster) to determine non-esterified fatty acids (NEFA), beta-hydroxybutyrate (BHB), glucose, cholesterol, calcium (Ca), phosphorus (P), magnesium (Mg), and alkaline phosphatase (AP) in serum, while aspartate aminotransferase (AST), glutamate dehydrogenase (GLDH) and gamma-glutamyltransferase (GGT) were analysed in heparin plasma. Glucose samples were collected in 6-mL fluoride plasma tubes (Vacuette; Greiner Bio-One GmbH, Kremsmünster) to avoid glycolysis during storage. After sampling, the serum samples were allowed to clot at room temperature for approximately 2 h, while plasma samples were stored at 7 °C before being separated by centrifuging at 3330 × *g* for 20 min. Thereafter, samples were stored at −20 °C, until further analyses. The concentrations of all parameters were measured at the laboratory of the Central Clinical Pathology Unit, University of Veterinary Medicine, Vienna. Standard enzymatic colourimetric analyses were conducted with a fully automated autoanalyzer for clinical chemistry (Cobas 6000/c501; Roche Diagnostics GmbH, Vienna, Austria). The intra-assay variation (the average coefficient of variation calculated from the individual coefficients of variation for all duplicates) was controlled by limiting the coefficient of variation to <5% for all blood variables.

### 2.8. Statistical Analysis

An ANOVA of feed intake, sorting, chewing, pH, blood, and milk data was performed using the mixed procedure of SAS (version 9.2, SAS Institute Inc., Cary, NC, USA). For each variable tested, the feeding phase (i.e., HS or MS wk 1, 2, 3, and 4) was considered as a fixed effect, while in the case of diurnal eating and ruminating the hour and hour × feeding phase were additionally considered. The individual cows were considered as random effects. Data obtained on the same cow but on different days within a feeding phase were considered as repeated measures with a first-order autoregressive variance–covariance structure. The significance level was set at *p* ≤ 0.05.

## 3. Results

### 3.1. DMI, Reticuloruminal pH, and Chewing Activity

The DMI, milk yield, and reticuloruminal pH of the cows are shown in Table 2.

Overall, DMI increased continuously during the experiment, being 2.5 kg higher during MS wk 4 (24.8 kg/d) as compared with HS (22.3 kg/d; *p* < 0.01). Furthermore, the variability of DMI was higher during MS (average standard deviation = 3.69) compared with HS (average standard deviation = 2.94). Feeding the MS diet led to a decline in reticuloruminal pH (min/d) below the SARA threshold from MS wk 1 onwards, while no further increase in the time of the pH < 6.0 (min/d) was observed during the entire MS period (*p* ≥ 0.06). However, a minor decrease of mean, minimum, and maximum pH (0.07, 0.06, and 0.13, respectively) was observed from MS wk 1 to 4 (*p* < 0.02). Variability of the mean pH increased from HS (0.17) to on average 0.24 during MS wk 1–3 (*p* < 0.01) but decreased afterward to 0.22 during MS wk 4 (*p* ≤ 0.01). 

While the DMI increased from MS wk 1 to 4 by 1.3 kg/d (*p* < 0.01), cows spent the same amount of time eating (on average 317 min/d) and total chewing (on average 856 min/d) during MS wk 1 and MS wk 4 (Table 3).

In contrast, the ruminating time during MS wk 4 was on an intermediate level as compared to the ruminating time during HS and MS wk 1. However, the differing ruminating time did not affect the number of ruminating chews per bolus, showing on average 60 ruminating chews per bolus irrespective of the feeding phase (*p* = 0.35). The chewing indices (eating, ruminating, and total chewing) related to the DMI (i.e., chews/g DMI and min/kg DMI) decreased linearly from HS to MS wk 4 (*p* ≤ 0.01), while chewing indices related to the NDF intake (i.e., eating, ruminating, total chewing/kg NDF, and peNDF > 8) showed their highest values during MS wk 1 (Table 3).

### 3.2. Daily Eating, Ruminating, and Mean pH Patterns

Daily eating and ruminating patterns, as well as the diurnal reticuloruminal pH dynamics, are shown in Figure 1.

The eating time was significantly affected by the feeding phase and time of the day, as well as their interaction (*p* < 0.01). Cows fed the MS diet reduced their eating time during the morning (0900–1000) and evening (1900–2000; *p* < 0.01) meals compared with cows fed the HS diet (Figure 1a). In addition, the morning eating time during MS wk 4 was further reduced as compared with MS wk 1 (*p* < 0.01). However, cows during MS showed more bouts of eating compared with HS before the morning and evening meal at 0500–0600, 0700–0800, and 1600–1800 (*p* ≤ 0.05). An additional meal after midnight (0100–0200) was observed during MS wk 4 as compared with MS wk 1 (*p* = 0.05), but not compared to HS (*p* = 0.54).

In accordance with the daily eating pattern, an effect of the feeding phase and time of the day, as well as an interactive effect between them, was observed for the daily ruminating time (*p* < 0.01). As shown in Figure 1b, cows increased ruminating directly after the main meals during MS wk 4. Thus, ruminating time increased by 7 min/h during MS wk 4 after the morning feeding, and by 5 min/h after the evening meal as compared with other feeding phases (*p* ≤ 0.05). However, cows during HS ruminated longer as compared with MS from 1100 to 1200 (*p* < 0.02), and as compared with MS wk 4 from 1600 to 1700 (*p* < 0.01).

Similar to the daily chewing patterns, daily mean pH differed slightly between all feeding phases (*p* < 0.01), with the highest values during HS (pH = 6.46), intermediate mean pH during MS wk 1 (pH = 6.35), and lowest values during MS wk 4 (pH = 6.28; *p* < 0.01). In addition, mean pH was affected by the time of the day (*p* < 0.01), with the strongest pH drops after morning feeding. Mean reticuloruminal pH was affected by the interaction of feeding phase and time of the day (*p* < 0.01), showing no difference between feeding phases 0100–0200. From 0300 (*p* < 0.01) on, pH decreased during MS wk 4 as compared with other feeding phases, while pH during MS wk 1 started to drop only after the morning meal at 0800 (*p* < 0.01). In addition, the mean pH during MS wk 4 was on average 0.09 pH units lower than during MS wk 1 after morning, afternoon, and evening feeding (until 1100, 1700, and 2000, respectively; *p* ≤ 0.05). 

Although the lowest pH values were observed during MS wk 4, diurnal variation of reticuloruminal pH was highest during MS wk 1 (SD = 0.065 vs 0.057 MS wk 4 and 0.055 HS; *p* < 0.01).

### 3.3. Sorting Behaviour

Actual feed refusals averaged 12% of the feed offered as fed over the course of the experiment and did not vary by feeding phase (*p* = 0.68). During the entire experiment cows sorted for medium and fine particles (1.18–19.0 mm; sorting index > 100), and against long particles (>19 mm) or very fine particles (<1.18 mm; sorting index < 100; Table 4). 

Comparisons among the feeding phases revealed that sorting behaviour against long particles (*p* = 0.04) and sorting in favour of medium particles (*p* < 0.01) was stronger during MS wk 4 as compared with MS wk 1 (Table 4). Further sorting against very fine particles increased from HS to MS wk 4 (*p* < 0.01). In contrast, sorting in favour of fine particles (1.18–8.0 mm) did not differ among the feeding phases. The actual intake of the particles mainly reflected the differences in the particle sizes between HS and MS, with cows fed the MS consuming fewer large particles and more medium and fine particles (*p* < 0.01; Table 4). However, the intake of very fine particles was decreased by 0.84 kg from MS wk 1 to 4 (*p* < 0.01).

### 3.4. Milk Yield, Milk Composition, and Feed Efficiency

Milk yield increased from 33.1 kg/d during HS to its maximum during MS wk 2 (35.2 kg/d), then dropped to 34.4 kg/d during MS wk 4 (*p* < 0.01; Table 2), while feed efficiency was highest during MS wk 3 (Table 5).

The milk fat content remained constant during the experiment (on average: 3.8%), while the protein content increased from 3.3 to 3.6% during the experiment (*p* < 0.01), resulting in a numerical decrease of the fat-to-protein ratio from 1.2 during HS to 1.0 during MS wk 4. While the milk fat yield was not affected by the feeding phase, the protein and lactose yields were higher during MS compared with HS (*p* < 0.01). Lactose concentration was slightly higher during HS than during MS (*p* = 0.01), while MUN dropped with the onset of MS-feeding by 6.4 mg/dl. During MS wk 4, MUN increased to reach the level of HS feeding. In addition, milk pH decreased slightly during MS and remained at a similar pH, averaging 6.6 from MS wk 1 to 4. The NDM ranged from 8.9% during MS wk 2 to 9.1% during MS wk 4, whereby only MS wk 4 differed from the HS feeding (8.9%). The SCC did not differ between feeding phases.

### 3.5. Blood Metabolites, Liver-Associated Variables, and Blood Minerals

Feeding phase effects on blood metabolites, blood minerals, and liver enzymes are shown in Table 6. Overall, blood metabolites related to the energy metabolism of the cows were altered when cows were switched from HS to MS, but remained constant during the entire MS feeding period. Plasma glucose concentration increased by 7.8 mg/dl, while BHB and NEFA decreased on average by 0.32 and 0.10 mmol/l respectively during MS as compared with HS (*p* < 0.01). Cholesterol decreased from HS to MS by an average of 18 mg/dl (*p* < 0.01), whereby only a numerical decrease from MS wk 2 to 4 was observed during MS (*p* = 0.10).

Determination of parameters related to the mineral metabolism revealed a decrease of Ca concentration by 0.1 mmol/l in MS wk 3 and 4 compared with HS (*p* = 0.03). While the highest concentrations of P were found in MS wk 4, a higher Mg level was found in MS wk 2, which returned to HS values in MS wk 4.

Overall, liver health-related variables showed the lowest values during HS and continuously increased with the duration of the MS feeding, exceeding the HS values from MS wk 2 (AP) or MS wk 3 (AST, GGT) onwards, respectively, while for GLDH a difference was only observed in MS wk 4. More specifically, AST increased by 58, GLDH by 48, and GGT by 4.9 U/l from HS to MS wk 4 (*p* < 0.01). Moreover, AP was elevated by feeding the MS diet by 5.0 U/l (*p* = 0.01) but did not differ among MS phases.

## 4. Discussion

The first aim of our study was to determine changes in the eating, sorting, and chewing behaviour of cows switched from a diet high in peNDF to moderate, and to examine associations between these variables and reticuloruminal buffering. The diet moderate in peNDF in our study contained approximately 18% peNDF_>8_ and 28% starch coming from barley grain. Besides being marginal in peNDF [4], this diet also was relatively high in starch, and most importantly the starch of barley is rapidly digestible in the rumen [16], making this diet an acidogenic diet. 

Feeding the acidogenic diet moderate in peNDF caused a significant decline in reticuloruminal pH below the SARA threshold, and cows altered chewing and sorting behaviour during MS feeding. These findings indicate that cows changed their sorting and chewing behaviour to mitigate the pH drop, but these changes were not enough to raise ruminal pH during MS feeding. Interestingly, the cows increased sorting for medium and against very fine particles during MS wk 4. Very fine particles, derived from starchy concentrate in the diet, are rapidly fermented and usually increase the risk of rapidly dropping ruminal pH, while longer fibre particles in the rumen promote stratification of the rumen content and saliva production [2]. Cows that increase their sorting for medium and against very fine particles during acidic rumen conditions, as observed in our study, have also been reported by DeVries et al. [5,18]. While cows sorted for long particles before and after SARA [18], the cows in our study sorted against long particles. One explanation for this might be that the proportion of long particles in our study was 39% higher than in the study by DeVries et al. [18], and high proportions of coarse particles increase the ability of cows to sort against them [2]. Moreover, a decreased palatability of long particles [14] likely enhanced the sorting against particles >19 mm. In contrast to the findings of DeVries et al. [5], cows continued sorting for fine particles containing concentrate during MS, assuming that cows did not reach their full potential in altering their sorting behaviour to recover from SARA during our experiment. Therefore, the decreased variability of reticuloruminal pH was likely sustained by other mechanisms mitigating variations in ruminal pH. For example, the smaller and more regular meals, as observed during MS wk 4, likely contributed to the reduced variability in ruminal pH. It has already been shown that smaller meals and a more evenly distributed feed intake during the day can improve the synchronization of volatile fatty acid (VFA) production and their elimination or neutralization [19]. In addition, cows enhanced ruminating behaviour after main meals, and the ruminating time during MS wk 4 ranged between the ruminating time during HS and MS wk 1, supporting the assumption of cows enhancing rumination over time to compensate for low reticuloruminal pH during high concentrate feeding [8]. The enhanced ruminating time might further be attributable to the cows increased sorting activity for medium and against very fine particles during MS wk 4, because the amount of medium and long particles in the diet are key factors for stimulating rumination [1]. However, it has to be mentioned that higher feed-intake levels typically promote longer daily chewing time, while decreasing the chewing time per kg DMI [20]. Therefore, the higher feed intake might have contributed to the intermediate ruminating time observed during MS wk 4. Ruminating enhances ruminal pH by stimulating saliva secretion because increased saliva secretion buffers the rumen content by direct neutralization of VFA [21,22]. Besides inducing cows to alter chewing and sorting behaviour to alleviate ruminal pH, feeding diets moderate in peNDF may also promote the growth of the ruminal papillae and adaptive events of ruminal microbiota [23]. These events could have resulted in an increased VFA and lactate uptake from the rumen when concentrate was fed, as shown previously [24,25], and likely helped in mitigating pH variation during MS wk 4. Overall, our results support the assumption of adaptive mechanisms, involving chewing and sorting behaviour to mitigate reticuloruminal pH variations, when cows are fed MS diets. However, our data suggest that activation of such adaptation mechanisms needs a certain time to evolve, as most effects were observed after feeding the MS diet for three weeks. In addition, these adaptive events cannot fully compensate for ruminal pH depression when diets provide only moderate peNDF levels.

A further aim of this study was to investigate the cows’ responses of chewing indices, milk and blood variables in a diet with a moderate peNDF level to establish their potential as rumen health indicators in dairy cattle. Several chewing thresholds have been suggested to be indicative of the sufficient physical effectiveness of diets (31 min total chewing/kg of DM) [26] or to avoid digestive disorders due to the feeding of diets low in physical structure (360 min ruminating/d) [27]. However, as chewing indices during MS even exceeded these thresholds, the chewing thresholds proposed earlier as indicators for the physical properties of diets are questionable. Another explanation might be that the decrease in ruminal pH in the current experiment was not severe enough to impair chewing behaviour.

Currently, the monitoring of milk fat and the fat-to-protein ratio are used as easily available indicators for adequate dietary fibre supply, and to detect fibre deficiency in dairy herds [3]. However, according to our results, the diagnostic value of the milk fat content in terms of predicting dietary conditions can be questioned. The milk fat content and yield were not affected by switching the cows from the HS to the MS diet, and the threshold of 3.4% milk fat, being indicative of milk fat depression during acidotic rumen conditions [28], was exceeded during the entire MS feeding period. Therefore, the milk fat content may stay in the normal range, although cows experience SARA. In contrast, the fat-to-protein ratio dropped to 1.0 in MS wk 4, an indicator pointing at acidotic rumen conditions or fibre deficient diets [3]. However, this threshold was reached only three weeks after cows experienced SARA, limiting its appropriateness as an early SARA indicator. During our experiment, the dropping fat-to-protein ratio can be attributed to the rising milk protein during MS, due to elevated rumen digestible organic matter and an increased microbial protein synthesis in the rumen [29]. Another finding, pointing to an increased reticuloruminal protein synthesis during low fibre feeding, is the dropping MUN level when switching the cows from HS to MS, as higher carbohydrate availability usually stimulates reticuloruminal protein synthesis and increases ammonia utilization, consequently decreasing MUN levels [30].

Paraclinical findings in blood variables during SARA may not be as distinct as in cases of acute clinical rumen acidosis, but varying degrees of compensated metabolic acidosis, hypocalcaemia, hyperphosphataemia, and increased liver enzymes may occur during SARA [3,31]. Indeed, our study revealed decreasing serum Ca during MS, as observed earlier when increasing the concentrate in the diet from 15% to 60% [32]. Diets were equal in Ca, and the increase of milk excretion of Ca should have been balanced for the increased Ca intake of cows fed MS diets. It is possible that feeding the MS diet may indirectly decrease serum Ca because of increasing LPS and VFA concentrations [33], or an increased inflammatory response during SARA [32,33]. However, Ca concentrations in our study remained within the physiological range [34], despite the MS diet. Therefore, using decreasing serum Ca to monitor cows with enhanced risk of SARA seems to be difficult. Additionally, lower concentrations of plasma cholesterol and higher glucose concentrations during high-concentrate feeding have been observed previously [35]. These changes seemed to be related to a higher absorption of glucogenic propionate at the expense of lipogenic acetate from the rumen [25,36]. Furthermore, decreased cholesterol during SARA might be linked to the translocation of LPS into the systemic circulation [37]. Moreover, we found liver enzymes exceeding reference values that indicated liver injury (GLDH 25, AST 105, and GGT 27 U/l) [38], when feeding the MS diet. This is an interesting finding, because liver-associated variables may increase when LPS and other toxins are translocated into the systemic circulation, causing liver cell damage [31], or it may be due to metabolic acidosis [3], inflammation, and oxidative stress [39,40,41]. These events were previously associated with the feeding of diets low in physical structure [26], or with acidotic rumen conditions [29,42], explaining why liver enzymes exceeded physiological thresholds during MS. In agreement with our results, liver health variables have previously shown to be elevated during SARA [31]. These findings indicate that liver health variables may help to identify cows experiencing SARA. However, a 4-wk high-grain feeding causing SARA did not lead to substantial alterations of liver health variables [35]. Therefore, further research investigating the exact relationship between feeding MS diets and liver health variables is warranted. Overall, increasing liver health variables and SARA induction during MS feeding emphasizes the importance of feeding diets meeting the peNDF > 8 mm recommendations, despite the improved energy balance and higher milk production during MS feeding.

## 5. Conclusions

Cows altered eating, sorting, and chewing behaviour when switched to a diet moderate in peNDF, and this likely contributed to reduced diurnal variations of reticuloruminal pH within the duration of concentrate feeding. However, feeding the MS diet induced SARA and had a negative impact on liver health variables, emphasizing the importance of peNDF level in mitigating SARA conditions. Investigation of multiple chewing, blood, and milk variables revealed that liver health variables exceeding physiological thresholds may help to identify cows experiencing SARA, while the informative value of the milk fat content and the ruminating chews per bolus seem to be limited in terms of predicting dietary or rumen conditions in cows. In conclusion, our study emphasizes that feeding diets with peNDF > 8 mm levels below 18% implies a SARA risk when their starch content, based on barley grain, exceeds 28% in DM basis.

## Figures and Tables

**Figure 1 animals-09-00128-f001:**
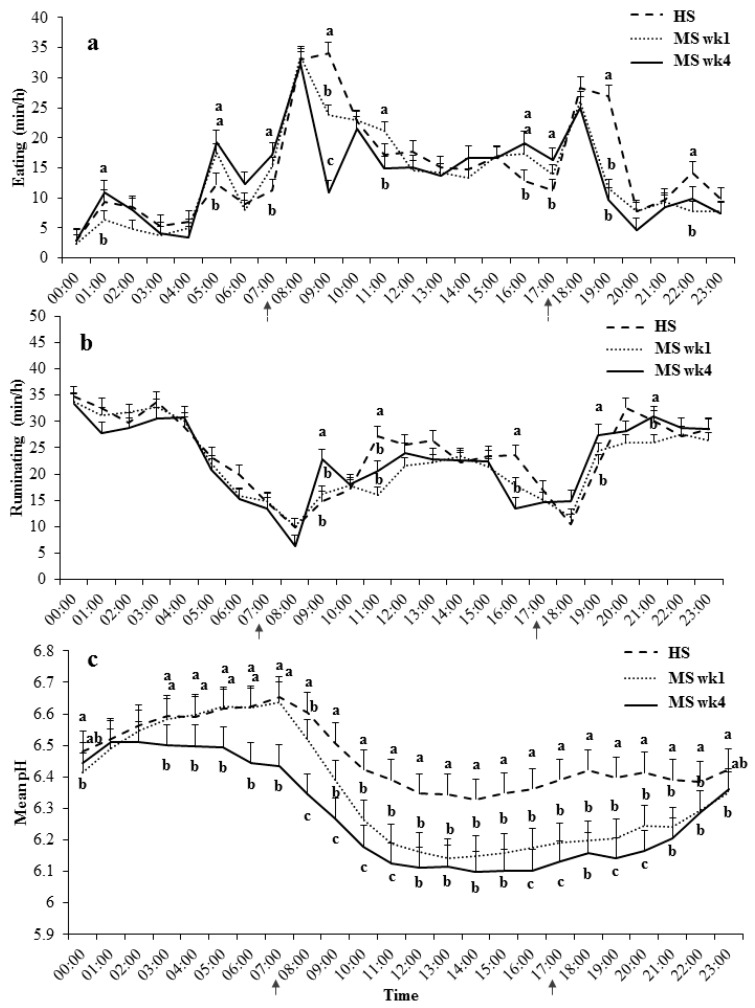
Daily eating (**a**), ruminating activity (**b**), and diurnal reticuloruminal pH dynamics (**c**) in cows fed a 40% concentrate diet with a high content of physical structure (24.4% peNDF > 8 mm DM basis; HS) for one week before being switched to a 60% concentrate diet with moderate physical structure content (17.6% peNDF > 8 mm DM basis; MS). Measurements were conducted during the last four days of HS until the end of MS wk 1, and on three consecutive days of MS wk 4. Arrows indicate times when the fresh feed was delivered (0730 and 1730). Hours with different superscripts differ significantly (*p* ≤ 0.05).

**Table 1 animals-09-00128-t001:** Ingredients, analysed nutrient composition, and particle size distribution in a 40% concentrate diet, high in physical structure (24.4% peNDF > 8 mm DM basis; HS) or a 60% concentrate diet with moderate physical structure content (17.6% peNDF > 8 mm; MS).

Item	HS	MS
Ingredient (% of DM)		
Grass silage	48.0	32.0
Meadow hay	12.0	8.0
Barley grain	25.2	37.8
Soybean meal	6.0	9.0
Corn	3.6	5.4
Rapeseed meal	3.2	4.8
Beet pulp	1.3	1.9
Mineral–vitamin premix ^1^	0.4	0.6
Monocalcium phosphate	0.2	0.3
Sodium chloride	0.1	0.2
Nutrient composition (% of DM unless otherwise stated)		
DM, % of fresh matter	46.1	47.6
Organic matter	91.0	92.3
Crude protein	16.2	17.2
Ether extract	2.1	2.5
aNDFom ^2^	40.0	32.5
ADF ^3^	24.1	18.6
Starch	18.8	27.7
NEL ^4^ (MJ/kg DM)	5.87	7.53
Particle size distribution (% of DM)		
>19 mm	45.5 ± 8.76	35.4 ± 4.83
8.0–19.0 mm	15.6 ±4.32	18.5 ± 8.52
1.18–8.0 mm	34.4 ± 4.67	41.6 ± 4.76
Pan (<1.18 mm)	4.5 ± 1.23	4.5 ± 1.16
pef ^5^ > 8	0.61	0.54
pef > 1.18	0.96	0.96
PeNDF ^6^ > 8 (% of DM)	24.4	17.6
peNDF > 1.18 (% of DM)	38.4	31.2

^1^ The mineral–vitamin premix contained 13.5% calcium, 9% magnesium, 5% phosphorus, 1.5% sodium, 2,100,000 IU vitamin A/kg, 300,00 IU vitamin D/kg, and 7,500 mg vitamin E/kg. ^2^ aNDFom, amylase-treated neutral detergent fibre. ^3^ ADF, acid detergent fibre. ^4^ Net energy for lactation (NEL) calculated according to the German Society of Nutrition Physiology (GfE) [10]. ^5^ pef, physically effective factor. ^6^ peNDF, physically effective neutral detergent fibre [11].

**Table 2 animals-09-00128-t002:** Dry matter intake (DMI), milk yield, and reticuloruminal pH in cows fed either a 40% concentrate diet high in physical structure (24.4% peNDF > 8 mm DM basis; HS) or a 60% concentrate diet with moderate physical structure content (17.6% peNDF > 8 mm; MS wk 1–4).

Item	Feeding Phase					SEM	*p*-Value
	HS	MS wk 1	MS wk 2	MS wk 3	MS wk 4		Phase
DMI (kg/d)	22.3 ^d^	23.6 ^c^	23.8 ^b,c^	24.6 ^a,b^	24.8 ^a^	0.65	<0.01
Milk yield (kg/d)	33.1 ^c,d^	34.0 ^b,c^	35.2 ^a^	35.1 ^a,b^	34.4 ^b^	1.1	<0.01
Time pH <6.0 (min/d)	182 ^b^	344 ^a^	308 ^a^	353 ^a^	358 ^a^	81.4	<0.01
Mean pH	6.46 ^a^	6.34 ^b^	6.33 ^b^	6.29 ^c^	6.27 ^c^	0.06	<0.01
Diurnal variation of pH ^1^	0.169 ^c^	0.239 ^a^	0.239 ^a^	0.243 ^a^	0.219 ^b^	0.0078	<0.01
Minimum pH	6.09 ^a^	5.87 ^b^	5.82 ^b,c^	5.79 ^c^	5.81 ^c^	0.06	<0.01
Maximum pH	6.81 ^a^	6.83 ^a^	6.78 ^b^	6.76 ^b^	6.70 ^c^	0.05	<0.01

^1^ Variability of pH indicates the standard deviation of diurnal pH per cow and feeding phase. ^a,b,c^ Means in the same row differ significantly among feeding phases (*p* ≤ 0.05).

**Table 3 animals-09-00128-t003:** Eating, ruminating, and total chewing behaviour in cows fed either a 40% concentrate die, high in physical structure (24.4% peNDF > 8 mm DM basis; HS), or a 60% concentrate diet with moderate physical structure content (17.6% peNDF > 8 mm; MS).

Item	Feeding Phase ^1^			SEM	*p*-Value
	HS	MS wk 1	MS wk 4		Phase
Eating					
Min/d	356 ^a^	320 ^b^	314 ^b^	16.8	<0.01
Chews/g of DMI ^2^	1.2 ^a^	1.0 ^b^	0.84 ^c^	0.07	<0.01
Min/kg DMI	16.4 ^a^	14.4 ^b^	12.4 ^c^	0.88	<0.01
Min/kg total NDF ^3^ intake	41.2 ^a,b^	44.0 ^a^	39.2 ^b^	2.57	0.10
Min/kg peNDF ^4^ _>8_	66.9 ^b^	86.1 ^a^	68.3 ^b^	4.42	<0.01
Min/kg peNDF_>1.18_	44.6	45.7	41.3	1.81	0.15
Ruminating					
Min/d	568 ^a^	532 ^b^	547 ^a,b^	11.9	0.02
Chews/g of DMI	1.7 ^a^	1.5 ^b^	1.3 ^c^	0.04	<0.01
Min/kg DMI	26.9 ^a^	23.7 ^b^	21.3 ^c^	0.67	<0.01
Min/kg total NDF intake	67.7 ^b^	72.6 ^a^	67.9 ^b^	2.04	0.01
Min/kg peNDF_>8_	110.0 ^c^	142.8 ^a^	118.1 ^b^	4.28	<0.01
Min/kg peNDF_>1.18_	74.2	74.2	72.0	2.18	0.68
Ruminating boli	597 ^a^	542 ^b^	557 ^b^	16.9	<0.01
Ruminating chews/bolus	59.0	59.8	60.2	1.42	0.35
Total chewing					
Min/d	926 ^a^	853 ^b^	860 ^b^	19.1	<0.01
Chews/g of DMI	2.9 ^a^	2.5 ^b^	2.2 ^c^	0.09	<0.01
Min/kg DMI	43.1 ^a^	38.1 ^b^	33.8 ^c^	1.20	<0.01
Min/kg total NDF intake	109 ^b^	117 ^a^	107 ^b^	3.54	<0.01
Min/kg peNDF_>8_	177 ^b^	229 ^a^	186 ^b^	6.62	<0.01
Min/kg peNDF_>1.18_	119	120	113	3.28	0.22
Total chews/min	70.6 ^a^	68.2 ^b^	67.9 ^b^	0.94	<0.01

^1^ Measurements were conducted during the last four days of HS, during MS wk 1 and on three consecutive days of MS wk 4. ^2^ DMI, dry matter intake. ^3^ NDF, aNDFom neutral detergent fibre assayed with a heat-stable amylase and expressed exclusive of residual ash. ^4^ peNDF, physically effective NDF _>8 or >1.18 mm_ [11]. ^a,b,c^ Means in the same row differ significantly among feeding phases (*p* ≤ 0.05).

**Table 4 animals-09-00128-t004:** Effects of the feeding phase on sorting behaviour of long, medium, short, and fine particles ^1^ in cows fed either a diet with 40% concentrate content high in physical structure (24.4% peNDF > 8 mm DM basis; HS), or a 60% concentrate diet with moderate physical structure content (17.6% peNDF > 8 mm; MS). Samples were taken during HS, MS wk 1 and during MS wk 4.

Item	Feeding Phase			SEM	*p*-Value
	HS	MS wk 1	MS wk 4		Phase
Sorting index ^1,^					
Long	95.2 ^a,b^	97.2 ^a^	90.0 ^b^	2.31	0.09
Medium	102.1 ^b^	101.5 ^b^	108.8 ^a^	1.69	0.01
Fine	105.3	102.4	105.2	1.08	0.12
Very fine	98.6 ^a^	84.9 ^a,b^	70.3 ^b^	6.18	0.01
Actual intake (kg/d) ^2^					
Long	20.3 ^a^	15.4 ^b^	16.7 ^b^	0.97	<0.01
Medium	8.75 ^b^	11.4 ^a^	13.5 ^a^	0.79	<0.01
Fine	17.3 ^b^	21.6 ^a^	21.1 ^a^	0.76	<0.01
Very fine	1.71 ^a^	1.91 ^a^	1.07 ^b^	0.15	<0.01

^1^ Particle size determined by Penn State Particle Separator, separating particles >19 mm (long), 8–19 mm (medium), 1.18–8 mm (fine), and <1.18 mm (very fine; Pan); ^2^ The sorting index (SI) was calculated as the ratio of actual intake to predicted intake for particles retained on each sieve of the separator. A sorting index >100 indicates sorting for particles, and a sorting index <100 indicates sorting against particles [14]. ^3^ The actual intake of each fraction was calculated as the difference between the amount of each fraction in the offered feed and that in the refused feed (in fresh basis). ^a,b^ Means in the same row differ significantly among feeding phases (*p* ≤ 0.05).

**Table 5 animals-09-00128-t005:** Effects of the feeding phase on milk composition in cows fed either a 40% concentrate diet high in physical structure (24.4% peNDF > 8 mm DM basis; HS), or a 60% concentrate diet with moderate physical structure content (17.6% peNDF > 8 mm; MS wk 2–4).

Item	Feeding Phase				SEM	*p*-Value
	HS	MS wk 2	MS wk 3	MS wk 4		Phase
ECM ^1^ (kg/d)	30.7 ^b^	33.4 ^a,b^	36.0 ^a^	32.9 ^a,b^	1.57	0.02
ECM/DMI ^2^ (kg/kg)	1.45 ^b^	1.37 ^b^	1.63 ^a^	1.37 ^b^	0.071	0.02
Fat (%)	3.9	3.7	3.9	3.6	0.21	0.38
Fat yield (kg/d)	1.22	1.27	1.48	1.24	0.099	0.12
Protein (%)	3.3 ^c^	3.5 ^b^	3.6 ^b^	3.6 ^a^	0.06	<0.01
Protein yield (kg/d)	1.05 ^b^	1.21 ^a^	1.23 ^a^	1.23 ^a^	0.040	<0.01
Fat: Protein	1.2	1.1	1.1	1.0	0.08	0.17
Lactose (%)	4.8 ^a^	4.7 ^b^	4.7 ^b^	4.7 ^b^	0.04	0.01
Lactose yield (kg/d)	1.51 ^b^	1.64 ^a^	1.64 ^a^	1.61 ^a^	0.059	<0.01
SCC ^3^ (log_10_/mL)	4.5	4.6	4.8	4.6	11.94	0.33
MUN ^4^ (mg/dl)	24.3 ^a^	17.9 ^c^	20.6 ^b^	23.3 ^a^	1.01	<0.01
pH	6.7 ^a^	6.6 ^b^	6.6 ^b^	6.6 ^b^	0.01	<0.01
NDM ^5^ (%)	8.9 ^b,c^	8.9 ^b^	9.0 ^a,b^	9.1 ^a^	0.08	<0.01

^1^ ECM, energy-corrected milk calculated as (0.38 × milk fat % + 0.21 × milk protein % + 0.95) × kg milk/3.2 [10]. ^2^ ECM/dry matter intake = feed efficiency. ^3^ SCC, somatic cell count. ^4^ MUN, milk urea nitrogen. ^5^ NDM, nonfat dry milk. ^a,b,c^ Means in the same row differ significantly among feeding phases (*p* ≤ 0.05).

**Table 6 animals-09-00128-t006:** Effects of the feeding phase on metabolites and blood minerals in cows fed either a 40% concentrate diet high in physical structure (24.4% peNDF > 8 mm DM basis; HS), or a 60% concentrate diet with moderate physical structure content (17.6% peNDF > 8 mm; MS wk 2–4).

Item	Feeding Phase				SEM	*p*-Value
	HS	MS wk 2	MS wk 3	MS wk 4		Phase
Glucose (mg/dl)	57.4 ^b^	66.1 ^a^	65.9 ^a^	63.5 ^a^	1.44	<0.01
Cholesterol (mg/dl)	198 ^a^	187 ^b^	178 ^b^	177 ^b^	8.32	<0.01
BHB ^1^ (mmol/l)	0.59 ^a^	0.34 ^b^	0.29 ^b^	0.33 ^b^	0.03	<0.01
NEFA ^2^ (mmol/l)	0.23 ^a^	0.10 ^b^	0.10 ^b^	0.09 ^b^	0.02	<0.01
Minerals ^3^						
Ca (mmol/l)	2.54 ^a^	2.47 ^a,b^	2.44 ^b^	2.44 ^b^	0.04	0.10
P (mmol/l)	1.39 ^b^	1.35 ^b^	1.45 ^b^	1.69 ^a^	0.09	0.01
Mg (mmol/l)	1.03 ^b^	1.16 ^a^	1.09 ^a,b^	1.08 ^b^	0.01	0.03
Liver health variables ^4^						
AST (U/L)	90.9 ^b^	105.4 ^b^	130.8 ^a^	149.3 ^a^	9.90	<0.01
GLDH (U/L)	17.5 ^b,c^	23.5 ^b^	43.7 ^a,b^	65.8 ^a^	9.85	<0.01
GGT (U/L)	25.4 ^c^	27.1 ^c,b^	28.1 ^b^	30.3 ^a^	1.60	<0.01
AP (U/L)	70.8 ^b^	75.6 ^a^	74.8 ^a^	77.1 ^a^	19.59	0.01

^1^ BHB, beta-hydroxybutyrate. ^2^ NEFA, non-esterified fatty acids. ^3^ Ca, calcium; P, phosphorus; Mg, magnesium. ^4^ AST, aspartate aminotransferase; GLDH, glutamate dehydrogenase; GGT, gamma-glutamyl transferase; AP, -alkaline phosphatase. ^a,b,c^ Means in the same row differ significantly among feeding phases (*p* ≤ 0.05).

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
