# Peer review of "Feeding Diets Moderate in Physically Effective Fibre Alters Eating and Feed Sorting Patterns without Improving Ruminal pH, but Impaired Liver Health in Dairy Cows"

_animals, 2019, doi:10.3390/ani9040128_

Round 1

Reviewer 1 Report

Dear authors,

The topic of your paper is of practical relevance with regard to threshold values for structure with maximum utilization of the starch content in the diet of dairy cows. Please find below a few comments and suggestions for correction.

Title

"....without improving ruminal pH and liver health in dairy cows" (nobody will expect an improvement in the health of the liver due to this scarce structural offer).

Proposal for a modification of the title:

"....without improving ruminal pH but impaired liver health in dairy cows"

Materials and Methods

line 81: information about DIM (days in milk) is missing; did you only use pluriparous cows for the study?

line 82: Which diet was previously fed (before starting with HS)? Did you allow the cows to get used to the new diet? When yes, how long was this period?

Table 1: The energy content of HS didn’t meet the requirement for the average milk yield (should be approx. 6.6 MJ NEL/kg DM assuming DMI of 23 kg).

line 107: Results didn’t show further information about weight development – delete

line 166: How did you calculate the mean pH? – mean [H+]-concentration- negative decadic logarithm – mean pH?

Results

line 268: Replace “MW” with “MS”

line 283: The diurnal variation of pH (standard deviation) is different than in table 2. This remains unclear.

line 305: Replace “Table 2” with “Table 4”

Discussion

Considering the low energy supply in HS, more changes in BHB and NEFA would have been expected. Do you have any explanation that the values are still in the reference range?

How do you explain the strongest drops in pH after morning feeding while after the evening feeding the pH is increasing in all groups?

References

Line 506: I can not find the appropriate reference: Do you mean: Proc. Soc. Nutr. Physiol. 2014; 23, 165-179?

Author Response

Reviewer 1:

Dear authors,
The topic of your paper is of practical relevance with regard to threshold values for structure with maximum utilization of the starch content in the diet of dairy cows. Please find below a few comments and suggestions for correction.
AU: Authors appreciate the affirmative review of Reviewer 1 and did their best to edit the paper according to the helpful comments.

Title
"....without improving ruminal pH and liver health in dairy cows" (nobody will expect an improvement in the health of the liver due to this scarce structural offer).
Proposal for a modification of the title:
"....without improving ruminal pH but impaired liver health in dairy cows"
AU: We edited the title accordingly

Materials and Methods
line 81: information about DIM (days in milk) is missing; did you only use pluriparous cows for the study?
AU: Sorry, our mistake. We added the respective information.

line 82: Which diet was previously fed (before starting with HS)? Did you allow the cows to get used to the new diet? When yes, how long was this period?
AU:
Before the start of the experiment cows were adapted to a diet with a maximum 40% concentrate for more than two weeks. We added the respective information.

Table 1: The energy content of HS didn’t meet the requirement for the average milk yield (should be approx. 6.6 MJ NEL/kg DM assuming DMI of 23 kg).
AU: Indeed, with the average milk yield the energy content of HS was not sufficient to fulfill the energy requirements, causing a negative energy balance of approximately 15 MJ NEL per day.

line 107: Results didn’t show further information about weight development – delete
AU: Done.

line 166: How did you calculate the mean pH? – mean [H+]-concentration- negative decadic logarithm – mean pH?
AU: The pH boli provided a pH measurement for every 10 min. For the diurnal analysis of the pH, the average of the 6 measurements per hour was used to obtain the mean pH values per hour and day. We edited this part of the material and methods section to improve comprehensibility.

Results
line 268: Replace “MW” with “MS”
AU: Sorry, our mistake. We corrected the abbreviation accordingly.

line 283: The diurnal variation of pH (standard deviation) is different than in table 2. This remains unclear.
AU: The data here derive from an additional analysis taking only the days where the chewing activity was measured into account. Therefore, the absolute values differ.

line 305: Replace “Table 2” with “Table 4”
AU: Done.

Discussion
Considering the low energy supply in HS, more changes in BHB and NEFA would have been expected. Do you have any explanation that the values are still in the reference range?
AU: During HS cows experienced an energy deficiency of about 15 MJ NEL per day, while the MS diet met (exceeded) the energy requirements, which was also reflected in decreased BHB and NEFA values in MS compared to HS. However, the energy deficiency of 15 MJ NEL per day during HS does not seem to overwhelm the capacity of the cow`s liver to process the mobilized body reserves. In early lactating cows more severe negative energy balances are commonly observed, as these cows also have a very limited DMI capacity, which predisposes them to excessive lipid mobilization (NEFA>0.7 mmol/L) and subclinical ketosis (BHB>1.4 mmol/L). In our study, NEFA and BHB remained below these thresholds in HS. In previous studies dairy cows experiencing similar NEB to the current study showed similar NEFA and BHB values as cows in the present study, remaining within reference values, too (e.g.
Kleefisch, MT; Zebeli, Q; Humer, E; Gruber, L; Klevenhusen, F(2018): Effects of feeding high-quality hay with graded amounts of concentrate on feed intake, performance and blood metabolites of cows in early lactation. Arch Anim Nutr. 2018; 72(4):290-307; Gruber, L; Khol-Parisini, A; Humer, E; Abdel-Raheem, SM; Zebeli, Q(2017): Long-term influence of feeding barley treated with lactic acid and heat on performance and energy balance in dairy cows. Archives of Animal Nutrition (71), 1 54-66.; Humer, E; Khol-Parisini, A; Gruber, L; Wittek, T; Aschenbach, JR; Zebeli, Q(2016): Metabolic adaptation and reticuloruminal pH in periparturient dairy cows experiencing different lipolysis early postpartum. Animal. 2016; 10(11):1829-1838).

How do you explain the strongest drops in pH after morning feeding while after the evening feeding the pH is increasing in all groups?
AU: This can be explained by the highest feed intake when the fresh feed was offered for the first time (morning feeding), which went along with a low ruminating activity during the daytime (i.e., from 5:00 to 18:00). On the other hand, feed intake decreased from 20:00 onward and ruminating activity increased from 19:00 onward. However, as this diurnal variation was observed in all feeding phases and the aim of our study was to highlight differences between HS and MS-feeding we do not want to go in more details on this aspect in the discussion.

References
Line 506: I can not find the appropriate reference: Do you mean: Proc. Soc. Nutr. Physiol. 2014; 23, 165-179?
AU: We edited the reference.

Reviewer 2 Report

Major comments:

The manuscript addresses an important topic to the dairy production sector, the effect of feeding diets moderate in physical effective fibre on eating, rumination, and feed sorting patterns as a strategy to regulate the rumen pH and energy intake. In addition, authors aimed to identify blood, milk and intake parameters that could be used as proxy to detect SARA at early stages. To achieve these aims, a feeding trial with 16 Simmetal dairy cows was performed. During one week cows were fed a high peNDF TMR and then were fed a moderate peNDF TMR for four weeks. In addition to the different peNDF levels, the TMR diets also had different F:C ratio and starch contents. Although authors discuss the peNDF effects on the several parameters, the F:C ratio and starch content are scarcely addressed or discussed. Additionally, it is not clear why the high and moderate peNDF TMR were fed for different periods (1 and 4 weeks). This should be clearly stated and justified, particularly as authors demonstrate that a 3 week period is needed for rumen parameters to stabilize after a diet change. Similarly, the data collection of the several parameters analyzed was not done in each week of the 5 week experiment, which may lead to biased conclusions. Another point that needs clarification is the statistical analysis. In face of the proposed aims and considering the experimental design, it is not clear why HS feeding phase is included in the model. Wouldn’t it made more sense to include the HS one week phase as covariable?

Minor comments:

L17: Revise the sentence “from a diet high to the diet (…)”

L68-70: Does you hypothesis considered parameters to be differently affected with time? Looking at the statistical analysis it seems to…

L72: Can you extrapolate the data obtained in Simmental cows to dairy cattle?

L82: Is one week enough? Check lines 407-409.

L87-88: Why the four week period?

L112: Why 0.5 mm samples? Except for starch, all chemical parameters should be determined in 1 mm samples (see AOAC methods). The analyses should be repeated in 1 mm samples to assure accuracy.

L158-161: What is the accuracy of the pH sensor?

L172-173: Check the sentence. It makes no sense on its own.

L174: “d 0 and 1 (HS)” Is this correct? This line makes no sense to what is stated in L124…

L196: Define what you mean by intra-assay variation.

L199-205: Why did you use feeding phase (wks) as fixed effects? Shouldn’t HS be covariable? Which are the random effects of the MIXED model?

L228: Table 3 should be placed in this line.

L264-265: But not with HS!

L305: Do you mean Table 4?

L327: Not true! The linear increase was only to week 3.

L332: MUN level in MS wk4 was the same as in HS, not almost.

L359-362: Only AP differed between HS and MS wk2!

L370: Nothing is expected in science!

L370-372: Was this due to peNDF or to the more acidogenic TMR?

L372-375: But their intake didn’t change…

L395-396: These medium particles promote rumination? Clarify.

L407-410: Wouldn’t a similar extent of time be needed for an adaptation to the HS diet?

L427: The ration was 1.0 in wk4 only. In wks 2 and 3 was 1.1.

L461-464: But MS diet induced SARA… Rephrase.

L468-470: But MS induced SARA. Rephrase.

L473-476: How can you conclude this if it was not evaluated?

Author Response

Reviewer 2:

Comments and Suggestions for Authors

Major comments:

The manuscript addresses an important topic to the dairy production sector, the effect of feeding diets moderate in physical effective fibre on eating, rumination, and feed sorting patterns as a strategy to regulate the rumen pH and energy intake. In addition, authors aimed to identify blood, milk and intake parameters that could be used as proxy to detect SARA at early stages. To achieve these aims, a feeding trial with 16 Simmetal dairy cows was performed. During one week cows were fed a high peNDF TMR and then were fed a moderate peNDF TMR for four weeks. In addition to the different peNDF levels, the TMR diets also had different F:C ratio and starch contents. Although authors discuss the peNDF effects on the several parameters, the F:C ratio and starch content are scarcely addressed or discussed. Additionally, it is not clear why the high and moderate peNDF TMR were fed for different periods (1 and 4 weeks). This should be clearly stated and justified, particularly as authors demonstrate that a 3 week period is needed for rumen parameters to stabilize after a diet change. Similarly, the data collection of the several parameters analyzed was not done in each week of the 5 week experiment, which may lead to biased conclusions. Another point that needs clarification is the statistical analysis. In face of the proposed aims and considering the experimental design, it is not clear why HS feeding phase is included in the model. Wouldn’t it made more sense to include the HS one week phase as covariable?

AU: Authors appreciate the affirmative evaluation of our manuscript. The comments were very helpful to improve the quality of our work.
We agree that the F:C ratio and starch content are important variables and the observed effects are not only related to the peNDF content then rather to the combination of these factors. Therefore, we point out that the diets differed in the F:C ration, and thus also in the peNDF and starch content.
Regarding the different duration of the feeding phases: Before the start of the experiment, cows were adapted to a diet with a maximum 40% concentrate for more than two weeks. As the HS diet was not sufficient to meet the energy requirements of the cows, we aimed to keep this period as short as possible to avoid any confounding effect of negative energy balance on the investigated parameters (such as damage to the liver due to excessive lipid mobilization already before feeding the MS-diets). Our main aim was to observe effects of a shift from a diet high in structure with a medium concentrate level to a diet marginal in structure with a high concentrate level and to investigate the effects of prolonged feeding of such diets, that are commonly fed to high-producing dairy cows. Therefore, we believe that demonstrating data from the HS-phase next to the MS-phases is more interesting for the reader (to enable direct comparisons on the effects of the dietary changes) than including the MS-phase as covariable.
We agree that more frequent sampling would have been beneficial. For the pH and DMI continuous data are available, while for other parameters such as the chewing data continuous monitoring was not feasible due to technical limitations.

Minor comments:

L17: Revise the sentence “from a diet high to the diet (…)”
AU: Done.

L68-70: Does you hypothesis considered parameters to be differently affected with time? Looking at the statistical analysis it seems to…
AU: Indeed, this was one major objective of our study, as this has not been investigated before (as described in the previous L58-60). We put more emphasis on this aspect in the edited introduction.

L72: Can you extrapolate the data obtained in Simmental cows to dairy cattle?
AU: Although Simmental cows are commonly considered as dual-purpose breeds, recent breeding programs focused stronger on milk production than on meat yield, as shown by the high milk yield of the cows used in the present study (about 35 kg/d). However, we cannot exclude possible differences among breeds, as chewing behavior, blood metabolites and milk composition are affected by various factors such as feed intake, farm management, illness, age and breed. However, authors investigating breed effects on chewing activity (Braun et al., 2015) or blood metabolites (Urdl et al. 2015) concluded that physiological responses of cows are primarily based upon differences in the feed (Braun et al. 2015) and energy balance (Urdl et al., 2015). Daily milk yield and milk composition may differ between cow breeds but studies comparing milk composition of different cow breeds found confounding results (Bendelja et al., 2011; Pintić et al., 2007; Johnson and Young, 2003; Abdouli et al., 2008), indicating that other factors (i.e. dietary effects and energy balance of the cow) are stronger than the breed-effect itself. The effect of different dietary peNDF levels on eating and feed sorting patterns, milk variables, chewing indices, blood minerals, and metabolites, as well as on liver enzymes can therefore likely be extrapolated to other dairy breeds and is not limited to Simmental cows.

References:
Abdouli et al. (2008): Non-nutritional factors associated with milk urea con- centrations under Mediterranean conditions. World Journal of Agricultural Sciences 4 (2):183-188.
Bendelja et al. (2011): Milk urea concentration in Holstein and Simmental cows (2011) Mljekarstvo 61 (1), 45-55.
Braun et al. (2015): Evaluation of eating and rumination behaviour in 300 cows of three different breeds using a noseband pressure sensor. BMC Veterinary Research 11:231.
Johnson and Young (2003): The association be- tween milk urea nitrigen and DHI production variables in Western commercial dairy herds. Journal of Dairy Science 86:3008-3015.
Urdl et al. (2015): Metabolic parameters and their relationship to energy balance in multiparous Simmental, Brown Swiss and Holstein cows in the periparturient period as influenced by energy supply pre- and post-calving. Journal of Animal Physiology and Animal Nutrition 99: 174–189.
Pintić et al. (2007): Kvantitativni pokazatelji kakvoće mli- jeka i hranidbeni status krava simentalske i holstein pas- mine Potkalničkog kraja. Krmiva 49 (2), 79-88.

L82: Is one week enough? Check lines 407-409.
AU: The HS-feeding was intended only to serve as a kind of reference for healthy rumen condition, while the major aim of our study was to reveal effects of switching to a diet marginal in structure and how the cows adapt to feeding this diets for a prolonged period. Moreover, one has to consider that before the start of the experiment cows were already adapted to a diet with about 40% concentrate. Therefore, one week was considered to be sufficient for our research objectives.

L87-88: Why the four week period?
AU: The MS-feeding was conducted for four weeks to observe the effects of prolonged feeding of a diet marginal in structure on feeding behavior, pH profile, liver health etc. and to investigate possible self-regulatory mechanisms of the cows to attenuate the fiber deficiency (as outlined in the introduction).

L112: Why 0.5 mm samples? Except for starch, all chemical parameters should be determined in 1 mm samples (see AOAC methods). The analyses should be repeated in 1 mm samples to assure accuracy.
AU: We agree that 1 mm seems to be sufficient for most chemical analyses. However, based on our experience, decreasing the screen to 0.5mm does not change the absolute values, but improves the reproducibility of the results (all analyses were conducted at least in duplicate). Furthermore, we confirmed our results with reference materials and therefore we do not see any need to repeat the analyses. Moreover, other labs use 0.5 mm screens for feed analysis as well (e.g. Haese et al., Journal of Animal Physiology and Animal Nutrition, 2017; 101(5):868-880).

L158-161: What is the accuracy of the pH sensor?
AU: We added the respective information.

L172-173: Check the sentence. It makes no sense on its own.
AU: Sorry, our mistake. We edited the respective sentence.

L174: “d 0 and 1 (HS)” Is this correct? This line makes no sense to what is stated in L124…
AU: d0 refers to the last day of the HS-feeding. To receive representative samples, milk samples were pooled from morning and afternoon milking. The samples on d1 were taken in the morning, before offering the MS-diet for the first time.

L196: Define what you mean by intra-assay variation.
AU: Done.

L199-205: Why did you use feeding phase (wks) as fixed effects? Shouldn’t HS be covariable? Which are the random effects of the MIXED model?
AU: As one major aim of our study was to investigate the effects of the switch from HS to MS, we prefer to keep HS in the model to show the respective values, than to solely include it as a covariable. Besides that, a further aim was to evaluate possible differences when the MS diet is fed for a prolonged time. Therefore, the different feeding phases (HS, MS wk1-4) were considered as fixed effects. We added the random effects to this section.

L228: Table 3 should be placed in this line.
AU: Done.

L264-265: But not with HS!
AU: We reported only the difference between MS wk1 and MS wk4 in that sentence, as HS was not differing from both phases. We edited the sentence according to the comment of the reviewer to highlight that no difference was observed compared to HS.

L305: Do you mean Table 4?
AU: Sorry, our mistake. We corrected accordingly.

L327: Not true! The linear increase was only to week 3.
AU: Sorry, our mistake. We corrected accordingly.

L332: MUN level in MS wk4 was the same as in HS, not almost.
AU: We referred to the absolute values (23.3 vs. 24.3). However, we agree that “almost” is not an appropriate term as this difference was not of statistical significance. We corrected accordingly.

L359-362: Only AP differed between HS and MS wk2!
AU: We edited this paragraph to highlight which liver enzymes were increased during which feeding phase compared to HS.

L370: Nothing is expected in science!
AU: Sorry, our mistake. We corrected accordingly.

L370-372: Was this due to peNDF or to the more acidogenic TMR?
AU: The low peNDF, together with the high concentrate/starch level was responsible for the acidogenic potential. Therefore, it is not possible to reduce to a single aspect. We edited the sentence.

L372-375: But their intake didn’t change…
AU: We agree that due to the increase in DMI from MSwk4 to MSwk1 the absolute intake of the long and medium particles did not change, although the sorting index changed. However, despite the higher DMI, the absolute intake of very fine particles was lower in MS wk 4 compared to MS wk1.

L395-396: These medium particles promote rumination? Clarify.
AU: According to the concept of peNDF (e.g. GfE, 2014), medium particles (>8mm) promote rumination activity, while very fine particles (<1.18 mm) do not. We put more emphasis on this aspect in the revised discussion.

L407-410: Wouldn’t a similar extent of time be needed for an adaptation to the HS diet?
AU: Before the start of the experiment cows were adapted to a diet with a maximum 40% concentrate for more than two weeks.

L427: The ratio was 1.0 in wk4 only. In wks 2 and 3 was 1.1.
AU: We specified accordingly.

L461-464: But MS diet induced SARA… Rephrase.

AU: Done.

L468-470: But MS induced SARA. Rephrase.
AU: Done.

L473-476: How can you conclude this if it was not evaluated?
AU: Thank you for this comment. We revised our conclusion.